# TUSCAN: Tumor segmentation and classification analysis in spatial transcriptomics

Chenxuan Zang[1], Charles C. Guo[2], Yaohong Wang[2], Peng Wei[1]*, Ziyi Li[1]*

**1** Department of Biostatistics, The University of Texas MD Anderson Cancer Center, Houston, Texas, United States of America, **2** Department of Pathology, The University of Texas MD Anderson Cancer Center, Houston, Texas, United States of America

* pwei2@mdanderson.org (PW); zli16@mdanderson.org (ZL)

## Abstract

The identification of tumor cells is pivotal for understanding tumor heterogeneity and the tumor microenvironment. Recent advances in spatially resolved transcriptomics (SRT) have revolutionized the way that transcriptomic profiles are characterized and have enabled the simultaneous quantification of transcript locations in intact tissue samples. SRT is a promising alternative method to study gene expression patterns in spatial domains. Nevertheless, the precise detection of tumor regions within intact tissue remains a great challenge. A common strategy for identifying tumor cells is via tumor-specific marker gene expression signatures, which are highly dependent on marker accuracy. Another effective approach is through aneuploid copy number alterations, as most types of cancer exhibit copy number abnormalities. Here, we introduce a novel computational method, called TUSCAN (TUmor Segmentation and Classification ANalysis in spatial transcriptomics), which constructs a spatial copy number variation profile to improve the accuracy of tumor region identification. TUSCAN combines gene information from SRT data and hematoxylin-and-eosin-staining image to annotate tumor sections and other benign tissues. We benchmark the performance of TUSCAN and several existing methods through the application to multiple datasets from different SRT platforms. We demonstrate that TUSCAN can effectively delineate tumor regions, with improved accuracy compared to other approaches. Additionally, the output of TUSCAN provides interpretable clonal evolution inferences that may lead to novel insights into disease development and potential druggable targets.

## Author summary

In our research, we addressed a fundamental challenge in understanding cancer: a tumor is not a single entity, but rather a complex and evolving ecosystem of different cells. To understand how cancer grows and becomes clinically aggressive, scientists need an accurate map of where cancerous cells are within

**Data availability statement:** All datasets analyzed in this study are publicly available. Detailed dataset information, including access links and descriptions, is provided in S1 Table. The code used in this study is publicly available at https://github.com/CZang409/TUSCAN.

**Funding:** This work was supported by the Cancer Prevention and Research Institute of Texas (CPRIT) (RP230166 to PW), the National Institutes of Health (NIH) (P50CA217674 and P01CA296429 to PW; R35GM159819 to CZ and ZL; U24CA274212 to ZL). The funders play no role in the study design, data collection and analysis, decision to publish, or preparation of the manuscript.

**Competing interests:** The authors have declared that no competing interests exist.

a tissue sample. Traditional methods for creating such maps often rely on gene markers that can be unreliable or vary from patient to patient, making it difficult to see the full picture. We developed a new computational tool called TUSCAN that instead looks for a more universal signature of cancer: copy number variations (CNVs), a hallmark of most tumors. By combining this genetic information with standard tissue images, TUSCAN can more accurately distinguish tumor areas from normal tissue. We tested our method across several datasets and found that it consistently outperformed existing tools. Beyond identifying tumor regions, TUSCAN also provides insights into how cancer cells evolve over time, offering a clearer view of tumor diversity. This could help researchers better understand the development of the disease and reveal new opportunities for treatment.

## Introduction

Intratumor heterogeneity, which is characterized by the presence of diverse tumor cell populations, is a driver of disease progression and represents a primary cause of therapeutic resistance. Understanding intratumor heterogeneity and the associated tumor microenvironment is crucial for uncovering the mechanisms behind cancer progression and invasion [1], as well as the modulation of immune cell activities [2]. Such insights not only enhance research into treatment responses [3,4], but also pave the way for more effective therapeutic interventions. In the past decade, single-cell technology has been successful in providing a comprehensive understanding of cell populations within the tumor microenvironment. Nonetheless, the absence of spatial information limits the ability of single-cell data to discern distinct tumor populations across various tissue locations.

Advances in spatially resolved transcriptomics (SRT) have provided a promising avenue to explore the spatial distribution of tumor cells [5,6]. One key advantage of SRT is its ability to preserve cell spatial locations during the generation of gene expression profiles. This unique feature enables a more profound exploration of gene expression patterns across various tumor regions in a spatial context, thereby allowing for the inference of tumor clonal substructures and tumor evolution analysis. The primary task in these analyses involves accurately pinpointing tumor regions within SRT data.

A conventional way to identify tumor cells is to use tumor-specific gene markers. For example, previous SRT studies performed initial clustering on SRT spots and defined cluster labels using known cell type markers [7]. A recently developed computational pipeline called TESLA [8] distinguishes tumors from other tissues through a machine learning algorithm and cancer gene markers [9]. However, this approach has several limitations. First, the efficacy of TESLA is contingent upon the selection of specific markers for different cancer types. This reliance introduces significant variability in the results, emphasizing the crucial role of marker accuracy. Second, the identification of consistent and reliable marker genes becomes extremely difficult in highly heterogeneous tumors such as triple-negative breast cancer [10] and

melanoma [11], which exhibit significant genomic variability. Tumor cells may even change their phenotype in response to treatment, rendering these markers not universally applicable to all individuals. These circumstances undermine the effectiveness of using typical gene markers for tumor identification.

An alternative approach to identifying tumor regions involves detecting aberrant copy number variations (CNVs), a hallmark present in the majority of cancer cells. Although copy number profiles (CNPs) originate from DNA-level alterations [12], several computational methods, including inferCNV [13], CopyKAT [14], and SCEVAN [15], have demonstrated the feasibility of inferring CNVs from single-cell RNA sequencing (scRNA-seq) data. These approaches rely on the principle that gene expression levels of adjacent genomic loci exhibit correlated patterns, enabling the estimation of CNPs through smoothing or aggregation across genomic regions. With the emergence of SRT techniques, inferCNV and related tools have been applied to spatial datasets [16–18]. For example, SpatialInferCNV [16] directly adapts the inferCNV workflow to SRT data. However, these approaches largely retain the modeling assumptions developed for scRNA-seq. As demonstrated in our experiments, SRT data exhibit distinct characteristics compared with scRNA-seq data, including spot-level aggregation and spatially correlated technical variation. Consequently, directly applying scRNA-seq-based CNV methods to SRT may attenuate biologically meaningful signals or introduce spatial inconsistencies in tumor region identification.

Given these challenges, it is necessary to develop an SRT-specific tool for distinguishing tumor regions from non-tumor tissues while overcoming the limitation of marker selections. Here, we introduce TUSCAN (**TU**mor **S**egmentation and **C**lassification **AN**alysis in spatial transcriptomics), an automated computational pipeline to perform tumor region segmentation and classification analysis by inferring spatial CNPs from gene expression and histology images in spatial transcriptomics data. Inspired by inferCNV [13], our method reconstructs CNPs by comparing the expression levels of genes in each spot to those in diploid spots, using a fixed-size sliding window across the genome to smooth the data. However, unlike inferCNV, our method can automatically select normal reference spots with high confidence using an evaluation score system that combines gene expression and histology images. TUSCAN also tailors the gene filtering, normalization, and normal residual signal neutralization procedures on the basis of the data characteristics from different SRT platforms. We applied TUSCAN to multiple SRT datasets, demonstrating its superior performance on tumor segmentation over existing methods that serve similar purposes. Furthermore, through a case study of human breast cancer SRT data, we illustrated how constructing spatial CNPs can facilitate tumor subclone classification and uncover distinct spatial CNV patterns among tumor subclusters. Moreover, visualization of CNV signals across spatial locations can aid in exploring cancer progression, understanding tumor lineage relationships, and identifying key mutational events that drive clonal differentiation. All these efforts offer deeper insight into intra-tumor heterogeneity. TUSCAN is available as user-friendly R software at https://github.com/CZang409/TUSCAN.

## Results

### Overview of TUSCAN

TUSCAN is applied to SRT data generated by sequencing-based platforms, including the original Spatial Transcriptomics platform [19] and its commercial successor, 10x Genomics Visium. We assume that both the SRT data and the associated hematoxylin-and-eosin (H&E)-stained images are available for analysis. Fig 1 shows the structured methodology of TUSCAN, which consists of three distinct steps. Step 1: Find a subset of spots most confidently classified as normal cells. In order to achieve this goal, we first divide all spots into several clusters, followed by the application of a Gaussian mixture model to estimate the gene expression variance within each cluster. Meanwhile, the histology image is transformed into a grayscale plot, from which the gray channel is extracted to compute the mean gray value for each cluster. Step 2: Infer CNVs. The histology image is used to identify normal tissue regions on the basis of two observations: first, the cluster exhibiting the minimum variance in gene expression usually indicates normal tissue, and second, the cluster with the lightest hue in the histology image is more likely to represent normal tissue than those with darker hues. Upon isolating a subset of normal spots, we use the mean gene expression value of these spots as a baseline reference for inferring the

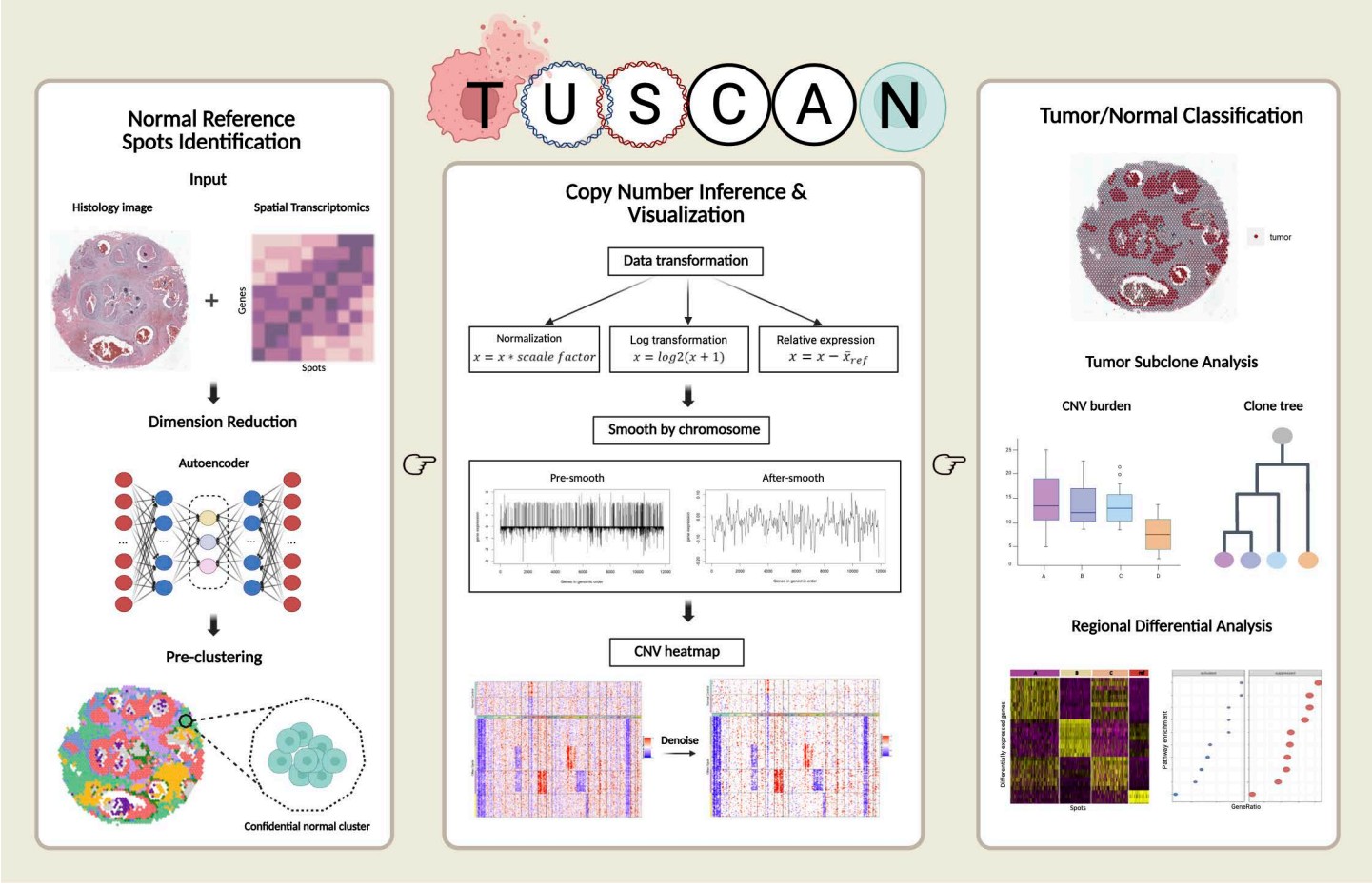

**Fig 1. Schematic overview of TUSCAN.** TUSCAN is designed to identify tumor regions in spatially resolved transcriptomics data. First, it integrates the histology image and spatial gene expression data as inputs; it then employs an autoencoder for dimensionality reduction. The low-dimensional features serve as the basis for an initial clustering of all spots. The cluster with the highest confidence of representing normal cells is defined (left box). A CNP for all spots is constructed (middle box) with the selected cluster as a normal reference. Finally, TUSCAN segments all spots into tumor regions and normal regions via consensus clustering with the CNP as the input. TUSCAN is also capable of performing a tumor subclone analysis, reconstructing a tumor clonal tree, and performing a regional differential analysis (right box). Created in BioRender. Zang, C. (2026) https://BioRender.com/gtpfhy9.

CNVs. This genome-wide CNV analysis is performed for all spots. Step 3: Identify the tumor region. The final step of TUSCAN employs consensus clustering to partition the tissue sample into two regions: tumor and normal tissue, predicted on the basis of the derived CNV matrix. A detailed description of these three steps is presented in the Methods section.

## Systematic evaluation of TUSCAN using four SRT datasets

We evaluated TUSCAN on six publicly available sequencing-based SRT datasets spanning multiple cancer types and platforms. In the main text, we present detailed results from four representative datasets. We first analyzed two human breast cancer datasets generated from the 10x Visium platform. The first sample was annotated with ductal carcinoma and invasive carcinoma, including 2,518 spots and 17,943 genes (Fig 2A) [20]. The second sample was annotated with ductal carcinoma in situ, lobular carcinoma in situ, and invasive carcinoma, including 3,798 spots and 36,601 genes (Fig 2B) [21]. The third dataset was a human invasive prostate cancer Visium dataset containing 4,371 spots and 17,943 genes (Fig 2C) [22]. The fourth dataset was a HER2-positive tumor sample profiled using the original Spatial

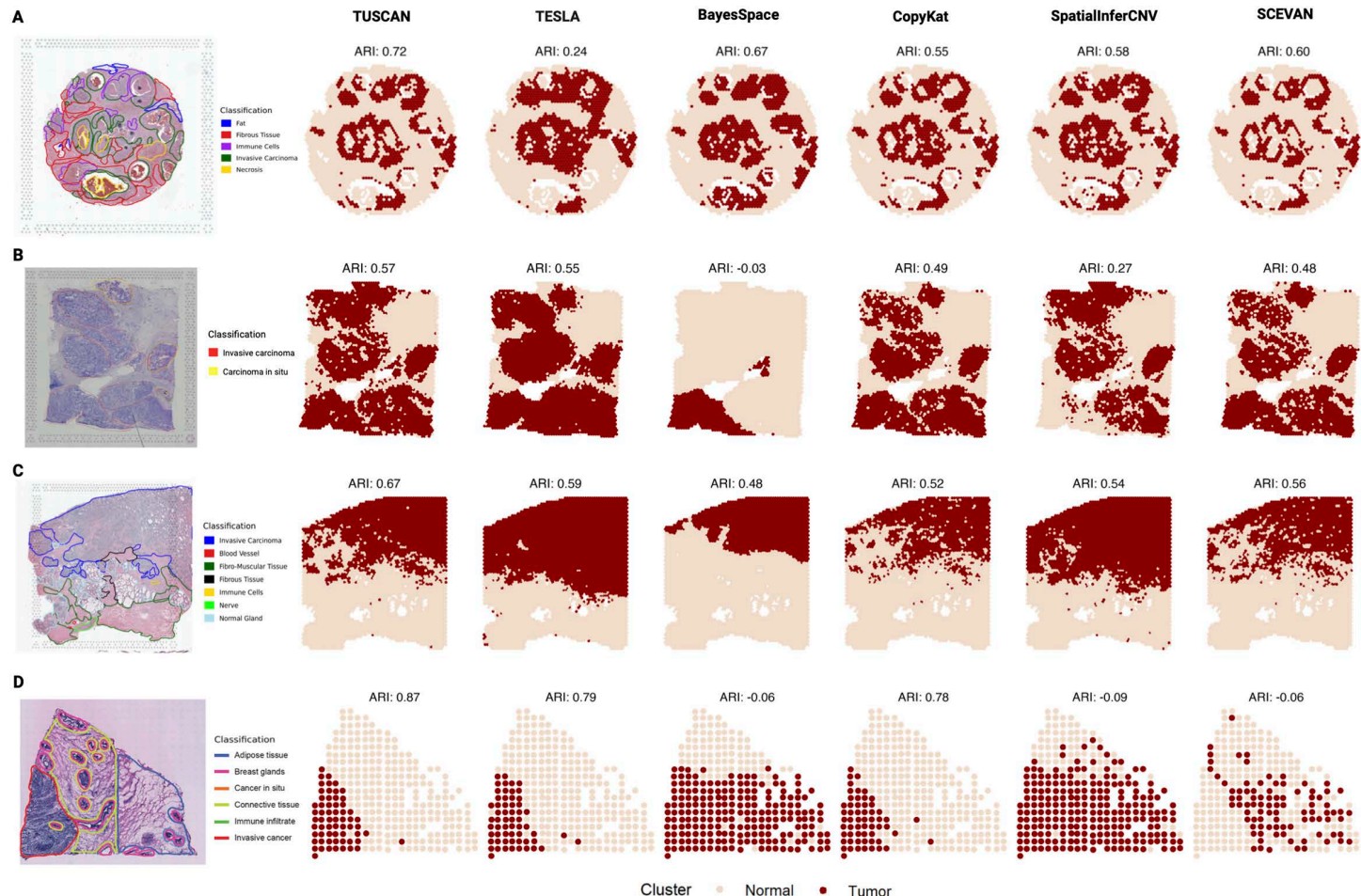

**Fig 2. Benchmark evaluation and comparison with other methods.** Pathologist-annotated histology images are in the left panel. Tumor sections of four human cancer samples identified by TUSCAN, TESLA, BayesSpace, CopyKat, SpatialInferCNV, and SCEVAN are marked in red. **(A)** Human breast cancer (Ductal Carcinoma in Situ). **(B)** Human breast cancer (Block A, Section 1). **(C)** Human prostate cancer. **(D)** Human epidermal growth factor receptor 2 + breast cancer, patient B.

Transcriptomics platform, including 293 spots and 16,148 genes detected (Fig 2D) [23]. To further demonstrate robustness and generalizability across cancer types, we additionally applied TUSCAN to a human primary pancreatic ductal adenocarcinoma (PDAC) Visium dataset (2,753 spots and 36,601 genes) and a human ovarian cancer Visium dataset (3,455 spots and 17,943 genes), whose results were shown in the Supplementary Materials. The corresponding results are presented in S3 and S4 Figs. Detailed dataset information is summarized in S1 Table.

We compared the tumor identification accuracy of TUSCAN with three categories of existing approaches: (1) TESLA [8], a marker-based method that identifies tumor regions based on spatially smoothed expression of predefined marker genes; (2) BayesSpace [24], a Bayesian spatial clustering framework that leverages neighborhood information for domain detection but does not explicitly model tumor-specific genomic alterations; and (3) CNV-based approaches, including inferCNV [13], CopyKAT [14], SCEVAN [15], and SpatialInferCNV [16], which infer large-scale copy number variation signals from transcriptomic data. Among CNV-based methods, inferCNV requires user-defined reference cells to establish baseline expression levels and does not automatically annotate tumor regions. When reference cells are not provided, it uses the average expression across all spots as the baseline. CopyKAT and SCEVAN implement internal strategies

to automatically identify putative normal cells and classify tumor versus normal populations. SpatialInferCNV adapts the inferCNV workflow to spatial transcriptomics data while retaining its reference-dependent framework. Tumor detection performance was evaluated using the Adjusted Rand Index (ARI) [25], measuring concordance between identified tumor domains and pathologist-annotated H&E regions.

Fig 2 demonstrates that TUSCAN outperforms existing methods in distinguishing tumor tissues, achieving the highest ARI values across all datasets. Notably, in the prostate cancer dataset, TUSCAN is the only method that attains an ARI value above 0.6. Beyond quantitative improvements, TUSCAN also exhibits a remarkable ability to precisely delineate tumor boundaries. In the human prostate cancer tissue, the tumor region consists of a predominant mass accompanied by several smaller adjacent foci. TUSCAN successfully separates and outlines these tumor regions, whereas TESLA tends to merge them into a single domain (Fig 2C). This highlights the advantage of jointly modeling CNV signals and spatial coherence for accurate tumor boundary detection. Overall, TESLA demonstrates competitive performance across datasets. However, its classification accuracy varies significantly across the three human breast cancer datasets when adopting a uniform set of marker genes to TESLA (Fig 2A, 2B and 2D). Specifically, its performance on the first dataset is notably inferior to that on the subsequent datasets, highlighting the inherent challenge of identifying consistent and reliable markers for cancers with high tumor heterogeneity, such as breast cancer, that are universally applicable to different individuals. In the human primary PDAC Visium dataset, TESLA could not be successfully executed (S2 Fig). This dataset contains two spatially disjoint tissue sections within the same slide. As the initial step of TESLA involves detecting tissue contours from the H&E image, we observed that TESLA identified only one of the tissue regions, leading to a runtime error in subsequent analyses. This behavior highlights a potential limitation of TESLA when applied to datasets containing multiple disconnected tissue regions. Furthermore, we observed that the performance of TESLA is highly sensitive to its built-in parameter $k$, which determines the number of genes used to calculate the relative expression and may present an obstacle for end-users.

CNV-based methods generally demonstrate commendable tumor identification performance across most Visium datasets. Also, it is worth noting that TUSCAN and other CNV-based methods could distinguish tumor from necrosis with remarkable precision (Fig 2A), whereas BayesSpace indiscriminately categorized necrosis as tumor, and TESLA failed to effectively differentiate between the two morphologies. This underscores the advantage of leveraging large-scale copy number signals for tumor detection.

Among the evaluated CNV-based tools, SCEVAN and CopyKAT exhibit relatively stable performance across the Visium datasets, with SCEVAN showing slightly superior accuracy overall. Notably, SCEVAN achieves the best performance on the human ovarian cancer dataset (S3 Fig). In contrast, SpatialInferCNV performs less consistently and yields lower accuracy compared to SCEVAN and CopyKAT on several Visium datasets. However, when applied to the ST platform dataset (HER2-positive breast cancer), both SpatialInferCNV and SCEVAN show substantially degraded performance and fail to clearly distinguish tumor from normal regions, suggesting potential cross-platform limitations when applied beyond their primary design context. Additionally, CopyKAT, SpatialInferCNV, and SCEVAN all exhibit a tendency to miss certain tumor spots, likely reflecting the challenges posed by the relatively shallow sequencing depth of SRT data compared to high-throughput scRNA-seq.

In contrast, BayesSpace shows the lowest accuracy among the evaluated methods, indicating that general-purpose spatial clustering approaches without explicit modeling of tumor-specific genomic alterations may lead to biased tumor identification.

## Case study: Human Breast Cancer (Block A, Section 1)

We conducted a comprehensive case study of the human breast cancer (Block A, Section 1) dataset [21]. The tissue sample was classified as AJCC/UICC Stage Group IIA, with estrogen receptor-positive, progesterone receptor-negative, and human epidermal growth factor receptor 2-positive. The examined sections contained ductal carcinoma in situ, lobular

carcinoma in situ, and invasive carcinoma. We first performed a tumor subclone analysis based on the CNPs computed by TUSCAN. We also constructed a tumor clonal tree to investigate tumor evolution and intratumor heterogeneity. Furthermore, we performed a differentially expressed gene (DEG) analysis and gene set enrichment analysis (GSEA) to identify distinct molecular patterns among tumor subclones.

**Clone substructure heterogeneity analysis.** The copy number matrix inferred from TUSCAN was extracted and clustered to identify potential tumor subpopulations. A total of 2265 tumor spots discerned by TUSCAN were divided into three main subclones (clones 1, 2, and 3). In Fig 3A, we illustrate the spatial distribution of these distinct tumor clones, mapped onto a histology image of the intact tissue. Each clone is confined to a well-defined region, indicating non-random spatial patterns with minimal intermingling between the subpopulations. The heatmap (Fig 3B) demonstrates that the three clones share certain similar CNV patterns, including copy number gains (1q, 8q, 17q) and copy number losses (7q, 11q). These genomic regions included many known breast cancer genes, for example, *MDM4*, *MYC*, *CCND1*, *ERBB2*, and *BRCA1* [26]. In addition to the common CNVs, each clone exhibits a distinct pattern. For example, clone 1 is characterized by a larger number of amplifications (3p, 5q, 12q, 16p, 20q) than clones 2 and 3 (Fig 3B and 3C). Clone 2 reveals increased amplifications on 1q and additional deletions on 17p and 22q. Clone 3 exhibits the most complex CNV patterns, which are discussed in more detail in the next section.

To further study the tumor heterogeneity, we conducted a DEG analysis and GSEA across these three tumor clones (Fig 3D, 3E and 3F). Clone 1 had a significant increase in the expression levels of several genes that are involved in immune cell recruitment, tumor-associated inflammation, angiogenesis, and proliferation, such as *CXCL14* and *CCND1* [27,28]. In clone 2, we observed a notable elevation of *CRISP3*, a gene that is closely linked to the immune response, indicating that it played a significant role in this clone's distinct genetic profile [29]. Our analysis of clone 3 highlighted an enhanced expression of key genes such as *CSTA*, *ERLIN2*, and *S100G*, which are crucial for preserving both the structural and functional integrity of cellular components [30–32].

Consistent with a more aggressive phenotype, GSEA (Fig 3F) revealed that clone 1 exhibits significant enrichment in genes associated with the epithelial–mesenchymal transition (EMT), coagulation, and the UV response (downregulated genes) compared to clones 2 and 3. The activation of the EMT program suggests that clone 1 may possess enhanced migratory and invasive capabilities, contributing to increased metastatic potential [33,34]. Furthermore, the enrichment of coagulation-related pathways points to a potential role in modulating the tumor microenvironment and promoting angiogenesis [35]. Conversely, clone 1 showed a marked suppression of Interferon-alpha (α) and Gamma (γ) signaling, indicating a diminished immune-response profile relative to clone 3 [36–38]. Additionally, both early and late estrogen response pathways were significantly downregulated in clones 1 and 2 compared to clone 3. Collectively, these GSEA signatures suggest that tumor clone 1 represents a more malignant subpopulation characterized by immune evasion, DNA damage response alterations, and heightened invasive potential.

**Tumor clonal tree reconstruction.** On the basis of the observations shown in Fig 3A and 3B, tumor clone 3 occupies the majority of the tissue sample and exhibits a complex CNV pattern. Consequently, we subdivided clone 3 into four subgroups based on its CNV profile and, along with clones 1 and 2, designated these as clones A through F (Fig 4A). Fig 4B presents the heatmap of CNPs for these six tumor clones. Although clones C–F (derived from clone 3) share similar CNV patterns, distinct differences are evident. We then constructed a phylogenetic tree, presented in Fig 4C, to illustrate the evolutionary trajectories of the six tumor clones on the basis of their CNV profiles, with gray nodes and one pink node representing the inferred unknown ancestral clones and original normal cells, respectively. Clones A and B exhibit distinct evolutionary paths as shown by the divergent branches originating from the normal cells, which may indicate that clones A and B do not originate from the same common clone as clones C to F. Clone D is the root within clone 3, demonstrating its status as an earlier-formed population within this cluster. According to the pathologist's annotations, it becomes evident that clone D essentially covers the carcinoma in situ region. The strong concordance between clone D and the pathologist-delineated carcinoma in situ area highlights the utility of clonal analysis in mapping tumor progression and

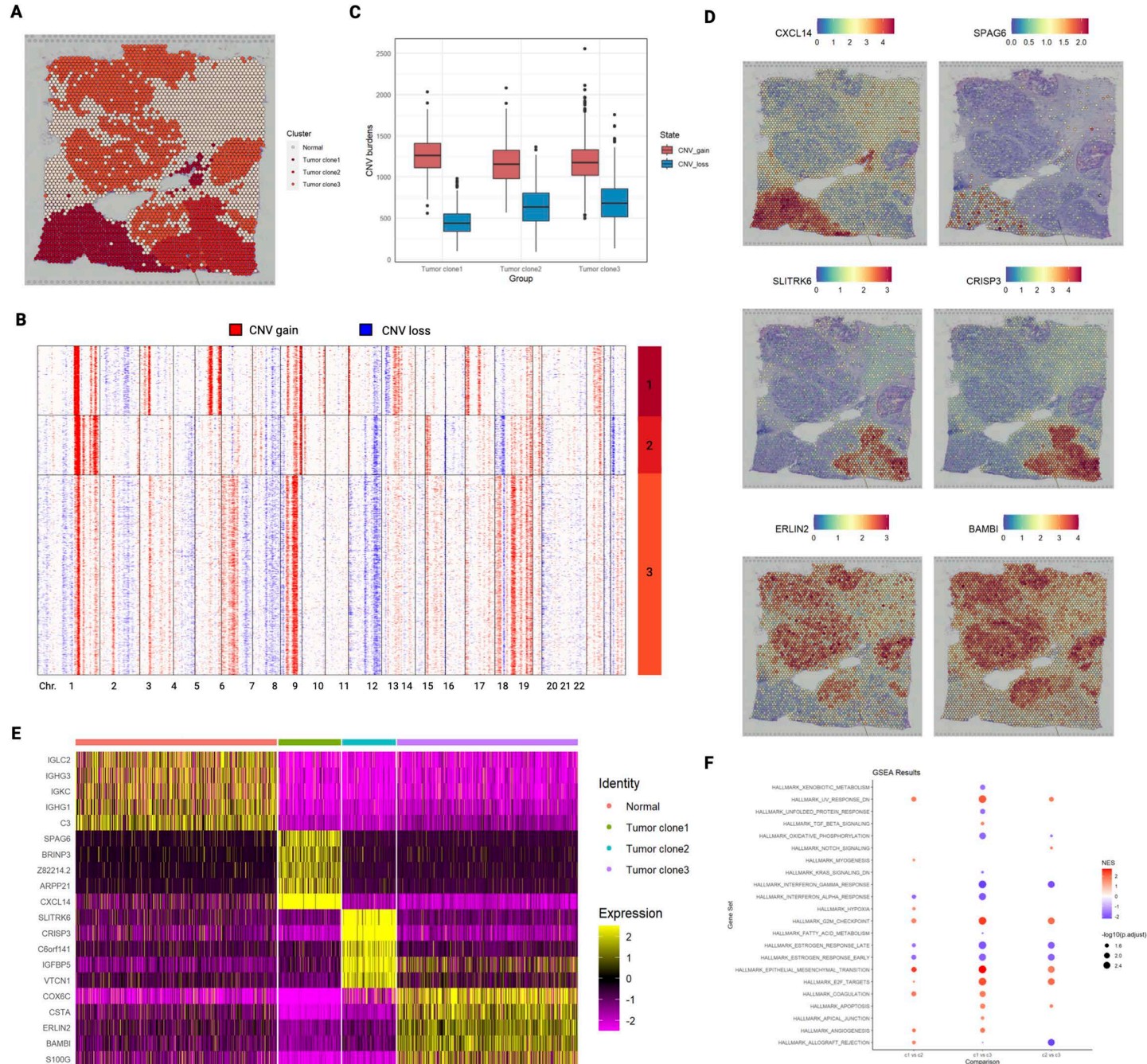

**Fig 3. Tumor clone analysis of human breast cancer. (A)** Spatial visualization of three tumor clones and normal tissue sections. **(B)** Heatmap of the estimated CNPs of tumor spots. Clonal classifications were determined by consensus clustering. **(C)** CNV burden calculated as the number of genes with extreme expression levels relative to the normal group. CNV_gain: $e_i > \text{mean}(e_{i,\text{normal}}) + 2\,\text{sd}(e_{i,\text{normal}})$. CNV_loss: $e_i < \text{mean}(e_{i,\text{normal}}) - 2\,\text{sd}(e_{i,\text{normal}})$. **(D)** Spatially variable features across three tumor clones. Top to bottom: clone 1 vs. clones 2 and 3; clone 2 vs. clones 1 and 3; clone 3 vs. clones 1 and 2. **(E)** Differential expression analysis of three tumor clones and normal cells. **(F)** GSEA of three tumor clones. Human hallmark gene sets from the Molecular Signatures Database (MSigDB) were used as the reference.

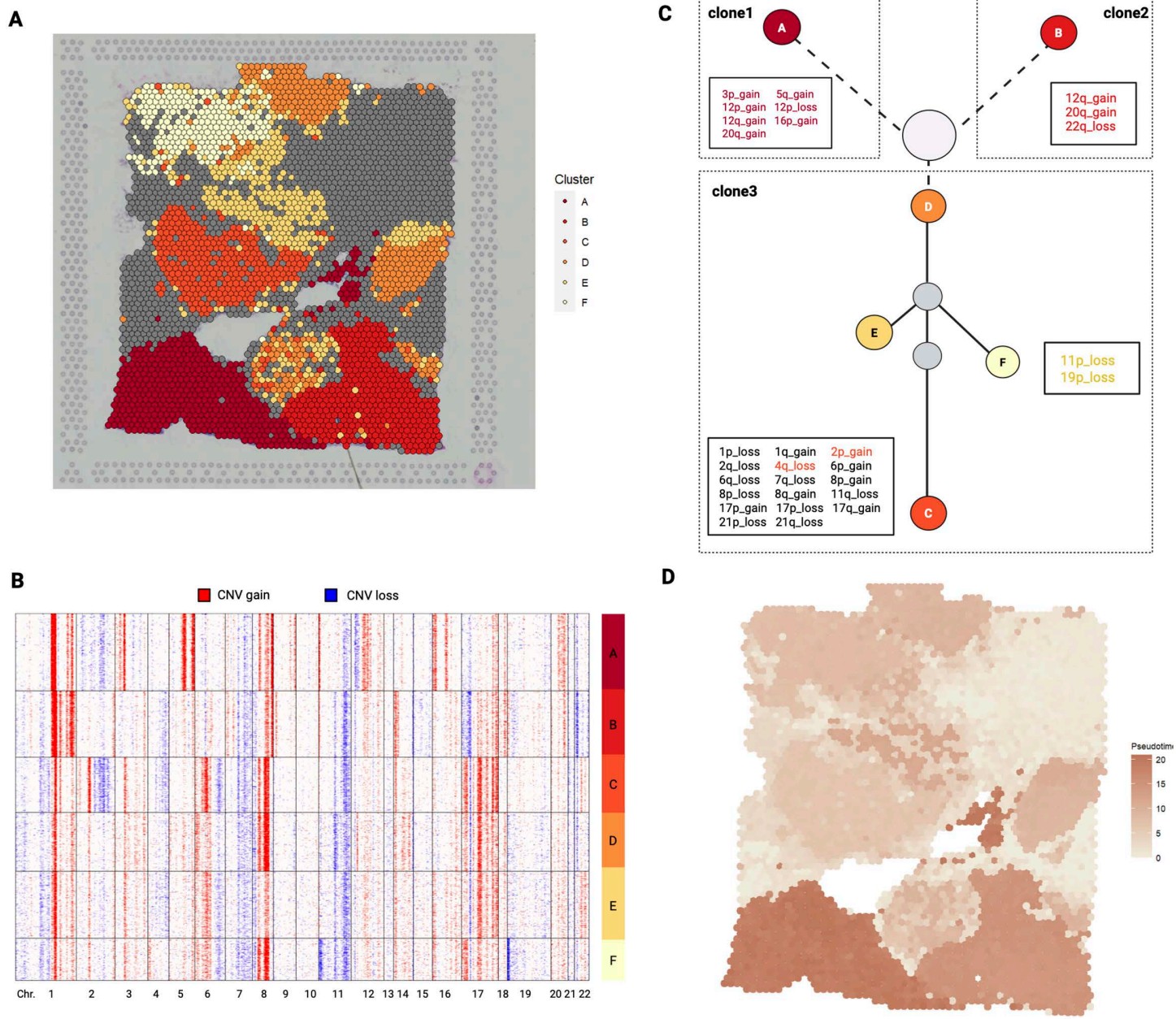

**Fig 4. Further subclone tree reconstruction. (A)** Spatial visualization of six tumor clones. **(B)** Heatmap of estimated CNPs for tumor spots. Clone 3 was further subdivided into four subclones. **(C)** Phylogenetic clonal tree of the tumor clones from inferred CNPs. Gray circles are unknown ancestors. The pink circle represents the original normal cells. All CNV locations of clone C are marked on the left side of the node, where the maximum CNV locations are observed. CNV locations that are unique to other clones in comparison to clone C are indicated with distinct colors. The complete CNV profile can be found in the S3 Table. **(D)** Visualization of the inferred trajectory. Higher pseudo-times are marked as darker colors.

identifying distinct tumor subregions. We performed trajectory analysis on tumor regions on the basis of CNPs generated from TUSCAN. The inferred pseudo-times indicate that clone 3 was formed earliest, while clones 1 and 2 were formed at a later stage (Fig 4D). These findings facilitate further examination of CNV loci within each subclone, enabling the discovery of distinct CNV patterns and the identification of specific gene markers across different tumor subclones.

## Discussion

In this paper, we present TUSCAN, a statistical method that integrates gene expression data and histology images to infer spatial copy number variations (CNVs) for tumor region segmentation. Through analyses of four real cancer datasets, we demonstrate that TUSCAN outperforms existing methods in accurately identifying tumor domains. A notable strength of our approach lies in its capability to autonomously delineate tumor regions, making it broadly applicable across a wide range of cancer types. This characteristic obviates the necessity for cancer-specific marker genes, thereby rendering our method particularly advantageous for cancers lacking well-established gene markers. The universality of TUSCAN not only facilitates its application but also significantly improves user accessibility and implementation across different cancer research contexts, aligning with the interdisciplinary nature of modern oncological studies.

By constructing spatial CNV profiles, we were able to distinctly delineate copy number amplifications and deletions within specific chromosomal bands in cancer samples. This allows for a deeper understanding of intratumoral heterogeneity and aids in the delineation of tumor subclones. In our breast cancer case study, we discerned three major subclusters (clones 1–3), which were further subdivided into six distinct subclones (clones A to F). By constructing a clonal tree and complementing it with trajectory analysis, we were able to delineate the evolutionary trajectory of the tumor. Our findings suggest that subclones C–F, which constitute clone 3, likely emerged at an early stage in the tumor's development. In contrast, subclones A and B (or clones 1 and 2) appeared at a later phase. Moreover, our comprehensive DEG analysis and GSEA results elucidated significant variations in gene signatures and pathways across the different tumor substructures. Notably, these analyses highlighted disparities in critical biological processes and pathways such as epithelial–mesenchymal transition, response to ultraviolet damage, coagulation, cellular responses to interferon alpha and gamma stimuli, and apoptosis.

We acknowledge several limitations of TUSCAN. First, owing to the inherent spot-based design of the technology, wherein each spot includes multiple cells (e.g., ST and 10x Visium), TUSCAN cannot achieve single-cell resolution. Consequently, the tumor spots identified through our method may not possess absolute purity. Second, TUSCAN might not be a suitable choice for analyzing cancer types where CNV events are rare or absent, such as chronic myeloid leukemia and acute myeloid leukemia [39,40]. Third, given the necessity of inferring the CNPs from the entire transcriptome, TUSCAN is not applicable to imaging-based SRT technologies, for example, MERFISH [41] and seqFISH [42], which are capable of simultaneously measuring only hundreds of genes. In future work, we will focus on integrating multi-omics data to refine resolution and enhance the precise tumor identification. Moreover, we plan to expand our efforts towards the annotation of a broader spectrum of cell types, enriching the depth and breadth of our analyses.

## Materials and Methods

### Tumor segmentation

**Step 1: Find a subset of high-quality normal cells.** The overall workflow of TUSCAN is shown in Fig 1. To accurately infer CNV from SRT data, we first need to identify a subset of normal spots with high confidence as the reference baseline values. We conduct a preliminary clustering on the basis of the spatial gene expression data and the H&E-stained image. The gene expression data is denoted by an $N \times D$ matrix of unique molecular identifiers where $N$ represents genes and $D$ represents spots. The raw gene expression matrix is normalized and log-transformed through the standard approach in the single-cell RNA data processing procedure. We then employ an autoencoder on the top 2000 highly variable genes to perform dimension reduction. By default, we obtain 20 dimensions, but users can also customize the number of dimensions after reduction. In addition to gene expression, we believe that the H&E-stained histology image provides invaluable insights for spatial clustering. Thus, for each spot $i$, we extract the three-channel RGB values by mapping each spot to the original histology image according to its 2D pixel coordinates $(x_i, y_i)$. Because the RGB intensity at a single pixel can be affected by local staining variation and pixel-level noise, we summarize local histological context by averaging RGB values within a square neighborhood centered at $(x_i, y_i)$. Specifically, for spot $i$, we compute

the mean RGB intensities within a $(2r + 1) \times (2r + 1)$ window and use these averages as $(r_i, g_i, b_i)$. The smoothing radius $r$ is selected based on a sensitivity analysis across multiple neighborhood sizes (S4–S7 Figs). Very small neighborhoods remained noisy, whereas moderate values preserved coherent tissue morphology while reducing pixel-level fluctuations. Across all datasets analyzed in this study, $r = 49$ (corresponding to a 99×99 window) provided a robust balance and was therefore used as the default setting. We note that spot diameters in the full-resolution image vary across datasets (typically 100–200 pixels in our data). Rather than directly defining the averaging region by the nominal spot diameter, we adopt a conservative central neighborhood to reduce potential boundary effects. Importantly, $r$ is an explicit user-adjustable parameter in TUSCAN. Next, the three color values are scaled to the same range as the 20 components, and these 23 features are used for preliminary clustering (Louvain or K-means). We recommend increasing the number of clusters while ensuring that the reference cluster contains at least 10 spots to reduce bias.

To determine the cluster with the highest potential purity of normal tissues, we create an evaluation score $s_c$ that combines the gene expression and histology image information as:

$$s_c = w_g g_c - w_\sigma \sigma_c. \tag{1}$$

Here, $g_c$ denotes the median gray value of cluster $c$. For each spot $i$, the gray value is computed as $g_i = 0.299 r_i + 0.587 g_i + 0.114 b_i$. The term $\sigma_c$ represents the standard deviation of gene expression within cluster $c$, estimated using a Gaussian mixture model.

To avoid one feature dominating the other, both $g_c$ and $\sigma_c$ are standardized before integration. The parameters $w_g$ and $w_\sigma$ control the relative contributions of gene expression and histological features, respectively. By default, we set $w_g = w_\sigma = 0.5$, corresponding to a symmetric baseline that assigns equal importance to transcriptional and imaging signals without imposing prior preference toward either modality. To evaluate the sensitivity of the method to the weighting scheme, we performed a systematic sensitivity analysis across a broad range of $w$ values (S8–S12 Figs). Across all datasets analyzed, the reference clusters selected under different weight settings were highly consistent and predominantly corresponded to non-tumor clusters. Moreover, downstream tumor segmentation accuracy, measured by ARI, remained consistently high across a wide range of $w$. These results indicate that TUSCAN is robust to moderate variations in the weighting parameter and that the final segmentation is not critically dependent on the default choice. We select the cluster with the maximum $s_c$ as the reference group on the basis of two observations. First, the cluster with the lightest color on the histology image is most likely to represent normal tissue. The rationale is that tumor cells may exhibit larger and more intensely stained nuclei due to an increased nuclear–cytoplasmic ratio, thus appearing darker than normal cells in the H&E-stained image. Second, the cluster with the lowest variance in gene expression is likely to represent normal tissue.

**Step 2: Infer the CNP.** Intuitively, in this step, we calculates a corrected moving average of gene expression data to determine CNV profiles. First, genes are sorted by absolute genomic position. Specifically, they are first ordered by chromosome and then by genomic start position within the chromosome. The underlying reason for this algorithm is that averaging the expression of genomically adjacent genes removes gene-specific expression variability and yields profiles that reflect chromosomal CNVs. To further refine the CNV profile of tumor cells, we construct the CNV profile of a known normal sample, and then for each gene and cell, the average normal reference is subtracted from each tumor spot to determine the final tumor CNV profile.

**Data preprocessing and transformation.** We first filter out those genes that have an average expression level of less than 0.01 in each spot and those that are detected in fewer than three spots. In the next step, we normalize the raw gene count matrix for each spot by the sum of unique molecular identifier (UMI) counts and multiply it by a scale factor (median unique molecular identifiers counts per spot). If the data are generated from the ST platform, we skip this step, as ST has a relatively low sequencing depth, and normalization would attenuate the CNV signal. We then perform a standard log-transformation on the data: $x = \log_2(x + 1)$.

**Obtain the relative gene expression values.** The gene expression values of all spots are subtracted by the average value of the normal reference. Since the data have been log-transformed, the resulting values represent the log-fold change differences relative to the control group. The outliers are handled as follows:

$$X = \begin{cases} x, & \text{if } |x| \le 3; \\ 3, & \text{if } x > 3; \\ -3, & \text{if } x < -3. \end{cases} \tag{2}$$

**Smooth along chromosome.** For a given gene $j$, the smoothed expression value $x_j$ is calculated as the weighted average of the genes within a window size $d$:

$$x_j = \sum_m w_m x_m, \tag{3}$$

where

$$w_m = \frac{n + 1 - m}{n + 1}, \quad n = \frac{d - 1}{2}.$$

Genes closer to the center gene $j$ are assigned larger weights. By default, the window size is set to $d = 101$.

**Further refine the CNV profile.** In the next step, the gene expression of each spot is centered by subtracting its median expression intensity. The average expression of the normal reference is subtracted again to further compensate for differences accrued after the smoothing process. The log transformation is reversed to symmetrically represent CNV gain or loss compared to neutral status.

**Reduce noise (optional).** To improve the signal-to-noise ratio and avoid over-interpreting minor transcriptional fluctuations, small deviations around the normal reference are suppressed. Specifically, gene expression values within 1.5 standard deviations of the normal mean are set to the corresponding normal mean:

$$X = \begin{cases} \mu_{\text{norm}}, & \text{if } |x - \mu_{\text{norm}}| \le 1.5\,\sigma_{\text{norm}}, \\ x, & \text{otherwise.} \end{cases} \tag{4}$$

This threshold is user-adjustable. By default, we use 1.5 standard deviations, following the established setting in the inferCNV pipeline, which has become widely adopted for balancing sensitivity and specificity in tumor–normal CNV comparisons.

**Step 3: Identify the tumor region.** We use a consensus clustering algorithm implemented in the R package '**ConsensusClusterPlus**' [43], to classify all spots into two clusters, tumor and normal, using the copy number matrix obtained from step 2 as input. The process is initiated by selecting a subset of both spots and genes from the copy number matrix. Each subset undergoes clustering into $k$ groups ($k = 2$ in our case), using a clustering algorithm such as hierarchical clustering, K-means, or a customized method. This clustering step is repeated multiple times.

Then pairwise consensus values, which are the frequency of two items being clustered together across various runs, are computed to construct a consensus matrix. The algorithm then performs a final round of agglomerative hierarchical consensus clustering using a distance measure based on 1 minus the consensus values, referred to as the consensus clusters. Finally, the distance is calculated between the centroids of two groups and that of the reference group. The group with the larger distance from the reference centroid is labeled as tumor.

We evaluate the performance of tumor segmentation using the Adjusted Rand Index (ARI) [25]. The ARI formula is defined as:

$$ARI(X, Y) = \frac{\sum_{ij} \binom{n_{ij}}{2} - \left[ \sum_i \binom{a_i}{2} \sum_j \binom{b_j}{2} \right] / \binom{n}{2}}{\frac{1}{2} \left[ \sum_i \binom{a_i}{2} + \sum_j \binom{b_j}{2} \right] - \left[ \sum_i \binom{a_i}{2} \sum_j \binom{b_j}{2} \right] / \binom{n}{2}},$$

(5)

where $X = \{x_1, x_2, \ldots, x_i\}$ is our spot label, $Y = \{y_1, y_2, \ldots, y_j\}$ is the true spot label, $n$ is the total number of spots, $n_{ij}$ is the number of spots in both cluster $X$ and cluster $Y$, $a_i$ is the total number of spots assigned in $x_i$, and $b_j$ is the total number of spots assigned in $y_j$. The range of ARI is from -1 to 1, with a higher ARI value indicating a better clustering result.

## Implementation details of existing methods

In applying TESLA to our datasets, cancer-type-specific marker genes were selected to construct the meta gene as recommended in the original TESLA framework.

For the three human breast cancer datasets, we used the 12 marker genes provided in the original TESLA publication as the breast cancer meta gene: *ERBB2*, *CNN1*, *CDH1*, *KRT5*, *KRT7*, *KRT14*, *KRT18*, *CDNND1*, *GATA3*, *FOXA1*, *PIP*, and *SCGB2A2* [8].

For the human prostate cancer dataset, nine marker genes were selected from existing literature to define the meta gene: *ADGRF1*, *CRISP3*, *TMEFF2*, *KCNC2*, *EGR*, *AMACR*, *OR51C1P*, *OR51E2*, and *MYO6* [16,44].

For the human primary pancreatic ductal adenocarcinoma (PDAC) dataset, four epithelial-associated marker genes were used to construct the meta gene: *ERBB2*, *GATA3*, *PIP*, and *SCGB2A2*.

For the human ovarian cancer dataset, five commonly reported epithelial and tumor-associated marker genes were selected: *EPCAM*, *KRT8*, *KRT18*, *PAX8*, and *MUC16*.

TESLA also requires an input parameter '*k*' to calculate the meta gene's relative expression at each superpixel, which is defined as the minimum value of the top *k* expressed genes from the meta gene list. Given that TESLA does not suggest how to determine the optimal value of *k*, we applied it with a range of *k* values to ascertain the most effective parameter.

CopyKat and SCEVAN were executed using the default settings. The classification results were obtained from the prediction result file. In the case of BayesSpace and SpatialInferCNV, due to their inability to directly label the tumor and other tissues, we first employed them to classify all spots into two groups, which were subsequently labeled based on the pathologist's annotations. SpatialInferCNV was implemented using the same reference cells as those used in TUSCAN. Other parameters were set to their default values.

## Tumor clone tree construction

Our tumor clone tree is constructed on the basis of the maximum-parsimony clone tree computed by the R package '**phangorn**' [45]. The 'user defined input' matrix to the R package phangorn is given in S2 Table. A detailed tutorial can be found in their vignettes titled 'Ancestral Sequence Reconstruction'. For the clone tree, the diameter of each node is proportional to the proportion of spots for this clone in the entire sample. The branch lengths are extracted from the tree object created by '**phangorn**'.

## Trajectory inference

As illustrated in Fig 4A, the tumor section is partitioned into six subclones, labeled A through F. We aim to delve deeper into the developmental trajectory of these tumor clones. For this purpose, we apply the R package '**Slingshot**' [46] to estimate the pseudotime associated with each individual spot. The CNV matrix serves as the initial input for this analysis, upon which UMAP dimensionality reduction is subsequently applied. Before the trajectory analysis, the normal cluster is

chosen as the start cluster. The pseudotime for each spot is extracted using the '**slingPseudotime()**' function and then projected onto the histology image, facilitating a comprehensive visualization of the tumor's evolutionary dynamics.

### DEG analysis and GSEA

A DEG analysis of the human breast cancer data is performed using '**Seurat v4**'. The significantly upregulated genes in each tumor clone compared to other clones and normal tissues are identified by the '**FindMarkers()**' function, with an adjusted p-value of 0.05. We further conduct a pairwise DEG analysis across all clones and perform GSEA on the detected DEGs using the '**GSEA()**' function in '**clusterProfiler**' [47] package employing default settings and an adjusted Benjamini-Hochberg p-value of 0.05. Human hallmark gene sets downloaded from the Molecular Signatures Database are used as the reference map.

### Supporting information

**S1 Fig. Effect of denoising on CNV inference in a human breast cancer (DCIS) Visium dataset.** Heatmaps display inferred copy number variation (CNV) profiles across genomic regions for Visium spots by TUSCAN. (A) CNV heatmap without denoising. (B) CNV heatmap after applying denoising, showing reduced noise and clearer CNV patterns.
(TIF)

**S2 Fig. Benchmark evaluation on additional human pancreatic cancer Visium dataset.** The left panel shows the pathologist-annotated H&E image of the human pancreatic cancer sample, where tumor regions are manually outlined in blue. The subsequent panels display the tumor regions (highlighted in red) as predicted by TUSCAN, BayesSpace, CopyKat, SpatialInferCNV, and SCEVAN. TESLA was excluded from this benchmark evaluation because it failed to complete. The presence of two spatially disjoint tissue fragments on a single Visium capture area (as shown in the H&E image) prevented TESLA's boundary-detection algorithm from accurately distinguishing between tissue and background, leading to execution failure.
(TIF)

**S3 Fig. Benchmark evaluation on additional human ovarian cancer Visium dataset.** The left panel shows the pathologist-annotated H&E image. Tumor regions predicted by TUSCAN, TESLA, BayesSpace, CopyKat, SpatialInferCNV, and SCEVAN are highlighted in red.
(TIF)

**S4 Fig. Effect of the smoothing parameter $r$ in TUSCAN on H&E-derived RGB features for human prostate cancer Visium dataset.** RGB values were extracted from the full-resolution H&E image at each spot location and locally averaged using a square window of size $(2r + 1) \times (2r + 1)$ pixel. The top-left panel shows the original H&E image. Remaining panels visualize spot-level RGB features under increasing $r$, illustrating how larger windows reduce pixel-level noise while preserving tissue morphology.
(TIF)

**S5 Fig. Effect of the smoothing parameter $r$ in TUSCAN on H&E-derived RGB features for human breast cancer Visium dataset (Ductal Carcinoma in Situ).** RGB values were extracted from the full-resolution H&E image at each spot location and locally averaged using a square window of size $(2r + 1) \times (2r + 1)$ pixel. The top-left panel shows the original H&E image. Remaining panels visualize spot-level RGB features under increasing $r$, illustrating how larger windows reduce pixel-level noise while preserving tissue morphology.
(TIF)

**S6 Fig. Effect of the smoothing parameter $r$ in TUSCAN on H&E-derived RGB features for human breast cancer Visium dataset (Block A, Section 1).** RGB values were extracted from the full-resolution H&E image at each spot location and locally averaged using a square window of size $(2r + 1) \times (2r + 1)$ pixel. The top-left panel shows the original H&E

image. Remaining panels visualize spot-level RGB features under increasing *r*, illustrating how larger windows reduce pixel-level noise while preserving tissue morphology.
(TIF)

**S7 Fig. Effect of the smoothing parameter *r* in TUSCAN on H&E-derived RGB features for human HER2-positive breast cancer ST dataset.** RGB values were extracted from the full-resolution H&E image at each spot location and locally averaged using a square window of size $(2r+1) \times (2r+1)$ pixel. The top-left panel shows the original H&E image. Remaining panels visualize spot-level RGB features under increasing *r*, illustrating how larger windows reduce pixel-level noise while preserving tissue morphology.
(TIF)

**S8 Fig. Effect of the weighting parameter *w* on reference cluster selection, human breast cancer Visium dataset (Ductal Carcinoma in Situ).** The top-left panel shows the clustering result for the dataset. The remaining panels illustrate the reference cluster selected under different values of *w* (as indicated), highlighting how varying w influences reference identification while preserving consistent spatial patterns.
(TIF)

**S9 Fig. Effect of the weighting parameter *w* on reference cluster selection, human breast cancer Visium dataset (Block A, Section 1) The top-left panel shows the clustering result for the dataset.** The remaining panels illustrate the reference cluster selected under different values of *w* (as indicated), highlighting how varying w influences reference identification while preserving consistent spatial patterns.
(TIF)

**S10 Fig. Effect of the weighting parameter *w* on reference cluster selection, human prostate cancer Visium dataset.** The top-left panel shows the clustering result for the dataset. The remaining panels illustrate the reference cluster selected under different values of *w* (as indicated), highlighting how varying w influences reference identification while preserving consistent spatial patterns.
(TIF)

**S11 Fig. Effect of the weighting parameter *w* on reference cluster selection, human HER2+ breast cancer ST Dataset.** The top-left panel shows the clustering result for the dataset. The remaining panels illustrate the reference cluster selected under different values of *w* (as indicated), highlighting how varying w influences reference identification while preserving consistent spatial patterns.
(TIF)

**S12 Fig. Robustness of ARI across weighting parameter *w* in four datasets.** (A) Human breast cancer (Ductal Carcinoma in Situ). (B) Human breast cancer (Block A, Section 1). (C) Human prostate cancer. (D) Human epidermal growth factor receptor 2+ breast cancer, patient B.
(TIF)

**S1 Table. Sources of published spatial transcriptomics datasets used in this study.**
(DOCX)

**S2 Table. File used to construct tumor clone tree of human breast cancer data (block A section 1).** This file summarizes the CNV state of each clone at different chromosome positions. The row names are all the chromosome bands where CNV appears, and the column names are clone A-F. We define five CNV states: 2 = more gain, 1 = less gain, 0 = neutral, -1 = less loss, -2 = more loss.
(XLSX)

**S3 Table. Chromosomal locations of copy number gains and losses of tumor clone A to F.**
(XLSX)

**S4 Table. Differentially expressed genes (clone 1 versus clone 2).**
(XLSX)

**S5 Table. Differentially expressed genes (clone 1 versus clone 3).**
(XLSX)

**S6 Table. Differentially expressed genes (clone 2 versus clone 3).**
(XLSX)

## Acknowledgments

We acknowledge the editing services provided by the Research Medical Library at The University of Texas MD Anderson Cancer Center.

## Author contributions

**Conceptualization:** Peng Wei, Ziyi Li.

**Data curation:** Chenxuan Zang, Charles C. Guo, Yaohong Wang.

**Formal analysis:** Chenxuan Zang.

**Funding acquisition:** Peng Wei.

**Investigation:** Chenxuan Zang, Peng Wei, Ziyi Li.

**Methodology:** Peng Wei, Ziyi Li.

**Software:** Chenxuan Zang.

**Validation:** Chenxuan Zang.

**Visualization:** Chenxuan Zang.

**Writing – original draft:** Chenxuan Zang, Charles C. Guo, Peng Wei, Ziyi Li.

**Writing – review & editing:** Chenxuan Zang, Charles C. Guo, Yaohong Wang, Peng Wei, Ziyi Li.

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
