## [Decision Letter · Decision Letter 0]

15 Dec 2025

PCOMPBIOL-D-25-02015

TUSCAN: Tumor segmentation and classification analysis in spatial transcriptomics

PLOS Computational Biology

Dear Dr. Li,

Thank you for submitting your manuscript to PLOS Computational Biology. After careful consideration, we feel that it has merit but does not fully meet PLOS Computational Biology's publication criteria as it currently stands. Therefore, we invite you to submit a revised version of the manuscript that addresses the points raised during the review process. You would need to assess performance of your method across a wider variety of tumor morphologies and image qualities as both reviewers mentioned the need.

We look forward to receiving your revised manuscript.

Kind regards,

Hatice Ulku Osmanbeyoglu, Ph.D

Academic Editor

PLOS Computational Biology

Ferhat Ay

Section Editor

PLOS Computational Biology

**Journal Requirements:**

2) Some material included in your submission may be copyrighted. According to PLOSu2019s copyright policy, authors who use figures or other material (e.g., graphics, clipart, maps) from another author or copyright holder must demonstrate or obtain permission to publish this material under the Creative Commons Attribution 4.0 International (CC BY 4.0) License used by PLOS journals. Please closely review the details of PLOSu2019s copyright requirements here: PLOS Licenses and Copyright. If you need to request permissions from a copyright holder, you may use PLOS's Copyright Content Permission form.

Potential Copyright Issues:

i) Figure 1. We note that the figure is created through BioRender. Please confirm that you hold a Premium account and provide a pdf copy of the CC BY 4.0 Licence as provided by BioRender. For instructions on how to generate a CC BY 4.0 license for your figure, please see the guidelines here: https://help.biorender.com/hc/en-gb/articles/21282341238045-Publishing-in-open-access-resources.

If you are using the free assets from BioRender, we are unable to publish these images as they are licenced under a stricter licence than CC BY 4.0. In this case we ask you to remove the BioRender images and replace them with open source alternatives.

See these open source resources you may use to replace images / clip-art:

- https://bioart.niaid.nih.gov/

- https://bioicons.com/

- https://healthicons.org/

- https://scidraw.io/

- https://reactome.org/icon-lib

- https://www.phylopic.org/images

- https://journals.plos.org/plosbiology/article?id=10.1371/journal.pbio.3002395

3) Please amend your detailed Financial Disclosure statement. This is published with the article. It must therefore be completed in full sentences and contain the exact wording you wish to be published.

4) Please revise your current Competing Interest statement to the standard "The authors have declared that no competing interests exist."

**Reviewers' comments:**

Reviewer's Responses to Questions

Reviewer #1: Summary

TUSCAN introduces an interesting approach by enabling automatic tumor identification in spatial transcriptomics without relying on pre-annotated tumor labels or marker genes. This is conceptually appealing and addresses limitations of existing methods, such as inferCNV, that require predefined tumor or normal references.

At the same time, there are some concerns regarding its robustness and general applicability:

1. Dependence on heuristics for normal reference selection

The method relies on selecting a normal reference cluster based on H&E brightness and gene expression variance. These heuristics may not consistently distinguish normal from tumor tissue, particularly in heterogeneous tissues, infiltrative tumors, or low-contrast images. Misidentifying the reference cluster could affect downstream CNV inference.

2. Aggressive data transformation

Replacing gene values within 1.5 standard deviations of the normal mean with the mean may reduce noise but could also smooth over real biological variation, potentially exaggerating CNV boundaries and tumor–normal differences.

3. Validation and robustness

While benchmarking against TESLA, CopyKAT, and necrosis delineation is helpful, broader testing across diverse tissue types and additional CNV inference methods would provide stronger evidence of robustness. More details on ground truth annotation and its uncertainty would also improve confidence in the evaluation.

Minor Issue and Concerns for the Author

1. The authors state that TUSCAN is applied to ‘spatial transcriptomics (ST) and 10x Visium’ data(L96-97). Technically, 10x Visium is a commercial implementation of the ST technology, and ‘spatial transcriptomics’ is a broader term referring to all spatially resolved transcriptomics methods. Clarifying this distinction would avoid potential confusion for readers

2. The justification for using a 99×99 pixel window for RGB averaging (L292) should be explained. In 10x Visium H&E images, the pixel size varies across datasets, and the footprint of a spot typically spans only ~10–30 pixels in many scans. pot size is determined by both spot_diameter_fullres and tissue_hires_scalef, and it varies between datasets. Using a fixed window of 99×99 patch may not be appropriate.

3. The weights for the H&E and gene expression score wg = wσ = 0.5.are set equally by default. The authors should briefly discuss how sensitive the final results are to changes in these weights.

Suggestions for the authors:

• Evaluate the sensitivity of tumor segmentation to normal reference selection and smoothing parameters.

• Benchmark against additional spatial CNV methods to clarify comparative performance.

• Provide more detailed ground truth validation and uncertainty estimates.

• Assess performance across a wider variety of tumor morphologies and image qualities.

Overall, TUSCAN presents a promising concept with a novel approach to automated tumor identification. However, additional validation and robustness analyses would strengthen confidence in its applicability across diverse datasets.

Reviewer #2: The manuscript introduces TUSCAN, a novel computational method for tumor segmentation in spatial transcriptomics data. Its core strength is the innovative use of spatial copy number variation profiles, which effectively bypasses the limitations of methods reliant on marker genes. The benchmarking results are compelling, showing that TUSCAN consistently outperforms existing methods.

The manuscript's detailed biological validation is confined to a single breast cancer case study, which may limit its appeal for the general Computational Biology journal. I suggest submitting to a special issue or a journal focused on cancer genomics, where this in-depth application would be a better fit.

**Have the authors made all data and (if applicable) computational code underlying the findings in their manuscript fully available?**

Reviewer #1: Yes

Reviewer #2: None

PLOS authors have the option to publish the peer review history of their article (what does this mean? ). If published, this will include your full peer review and any attached files.

**Do you want your identity to be public for this peer review?** For information about this choice, including consent withdrawal, please see our Privacy Policy .

Reviewer #1: No

Reviewer #2: No

**Figure resubmission:**
---

## [Decision Letter · Decision Letter 1]

24 Feb 2026

Dear Dr Li,

We are pleased to inform you that your manuscript 'TUSCAN: Tumor segmentation and classification analysis in spatial transcriptomics' has been provisionally accepted for publication in PLOS Computational Biology.

Best regards,

Hatice Ulku Osmanbeyoglu, Ph.D

Academic Editor

PLOS Computational Biology

Ferhat Ay

Section Editor

PLOS Computational Biology

Reviewer's Responses to Questions

**Comments to the Authors:**

Reviewer #1: The authors have satisfactorily addressed my previous concerns. The manuscript has improved and I recommend acceptance.

Reviewer #2: The authors have addressed all the concerns, particularly by including additional examples to validate the claims. The revised manuscript reads very sound

**Have the authors made all data and (if applicable) computational code underlying the findings in their manuscript fully available?**

Reviewer #1: None

Reviewer #2: Yes

PLOS authors have the option to publish the peer review history of their article (what does this mean? ). If published, this will include your full peer review and any attached files.

**Do you want your identity to be public for this peer review?** For information about this choice, including consent withdrawal, please see our Privacy Policy .

Reviewer #1: **Yes:** Xiaojun Ma

Reviewer #2: No

---

## [Editor Report · Acceptance letter]

PCOMPBIOL-D-25-02015R1

TUSCAN: Tumor segmentation and classification analysis in spatial transcriptomics

Dear Dr Li,

I am pleased to inform you that your manuscript has been formally accepted for publication in PLOS Computational Biology. Your manuscript is now with our production department and you will be notified of the publication date in due course.

With kind regards,

Anita Estes
